# The Role of Cardiac N-Methyl-D-Aspartate Receptors in Heart Conditioning—Effects on Heart Function and Oxidative Stress

**DOI:** 10.3390/biom10071065

**Published:** 2020-07-16

**Authors:** Natalia Govoruskina, Vladimir Jakovljevic, Vladimir Zivkovic, Isidora Milosavljevic, Jovana Jeremic, Jovana Bradic, Sergey Bolevich, Israpil Alisultanovich Omarov, Dragan Djuric, Katarina Radonjic, Marijana Andjic, Nevena Draginic, Aleksandra Stojanovic, Ivan Srejovic

**Affiliations:** 1Department of Human Pathophysiology, I.M. Sechenov First Moscow State Medical University, Trubetskaya str. 2, 119992 Moscow, Russia; natalia.govorushkina@gmail.com; 2Department of Physiology, Faculty of Medical Sciences, University of Kragujevac, Svetozara Markovica 69, 34000 Kragujevac, Serbia; drvladakgbg@yahoo.com (V.J.); vladimirziv@gmail.com (V.Z.); 3 Laboratory of Navigational Redox Lipidomics, Department of Human Pathology, I.M. Sechenov First Moscow State Medical University, Trubetskaya str. 2, 119992 Moscow, Russia; bolevich2011@yandex.ru; 4Department of Pharmacy, Faculty of Medical Sciences, University of Kragujevac, Svetozara Markovica 69, 34000 Kragujevac, Sebia; isidora.stojic@medf.kg.ac.rs (I.M.); jovana.ilona@gmail.com (J.J.); jovanabradickg@gmail.com (J.B.); katarina.radonjic@medf.kg.ac.rs (K.R.); andjicmarijana10@gmail.com (M.A.); nevenasdraginic@gmail.com (N.D.); vranicaleksandra90@gmail.com (A.S.); 5Medical and Health Center of the Ministry of Foreign Affairs of Russia, Smolensky b-r, 32/34, 119002 Moscow, Russia; omarov-ia@mail.ru; 6Institute of Medical Physiology “Richard Burian”, Faculty of Medicine, University of Belgrade, str. Visegradska 26/2, 11000 Belgrade, Serbia; drdjuric@eunet.rs

**Keywords:** N-methyl-D-aspartate receptor, glutamate, heart, conditioning, cardiodynamics, oxidative stress

## Abstract

As well as the most known role of N-methyl-D-aspartate receptors (NMDARs) in the nervous system, there is a plethora of evidence that NMDARs are also present in the cardiovascular system where they participate in various physiological processes, as well as pathological conditions. The aim of this study was to assess the effects of preconditioning and postconditioning of isolated rat heart with NMDAR agonists and antagonists on heart function and release of oxidative stress biomarkers. The hearts of male Wistar albino rats were subjected to global ischemia for 20 min, followed by 30 min of reperfusion, using the Langendorff technique, and cardiodynamic parameters were determined during the subsequent preconditioning with the NMDAR agonists glutamate (100 µmol/L) and (RS)-(Tetrazol-5-yl)glycine (5 μmol/L) and the NMDAR antagonists memantine (100 μmol/L) and MK-801 (30 μmol/L). In the postconditioning group, the hearts were perfused with the same dose of drugs during the first 3 min of reperfusion. The oxidative stress biomarkers were determined spectrophotometrically in samples of coronary venous effluent. The NMDAR antagonists, especially MK-801, applied in postconditioning had a marked antioxidative effect with a most pronounced protective effect. The results from this study suggest that NMDARs could be a potential therapeutic target in the prevention and treatment of ischemic and reperfusion injury of the heart.

## 1. Introduction

The N-methyl-D-aspartate receptors (NMDARs) belong to the ionotropic glutamate receptor family. Structurally, NMDAR is a tetrameric complex composed of GluN1 and GluN3 subunits, which bind glycine (or D-serine), and glutamate-binding GluN2 subunits, which combine differently in the assembly of NMDARs [1]. There are eight splice variants of GluN1, four different GluN2 subunits (A–D) and two GluN3 subunits (A and B). Various compositions of NMDAR assembly result in different biophysical, pharmacological and functional features of NMDARs in distinct regions of the body [2,3]. NMDARs have several unique features, which makes them completely different from other glutamate receptors: (1) NMDARs require simultaneous binding of both co-agonists, glutamate and glycine, for activation, (2) the NMDAR pore is blocked by Mg^2+^, which is removed upon membrane depolarization (characteristic of voltage-gated channels), and (3) activated NMDARs are highly transient to Ca^2+^, unlike other glutamate receptors which are mainly Na^+^ channels [1,4]. The high permeability of NMDARs for Ca^2+^ results in considerable changes in intracellular Ca^2+^ content upon modulation of NMDAR activity, which consequently affects Ca^2+^-mediated intracellular cascades [5,6]. Beside the fact that the primary expression and the most prominent role of NMDARs are related to the function of the central nervous system (CNS), there is growing interest in the role of NMDARs in other extraneural organs and tissues, including cells and tissues within the cardiovascular system [7]. The results of several studies regarding the presence of different NMDAR subunits in cardiovascular tissues have led to contradictory conclusions. Firstly, the heart expression of several forms of GluN2 subunits was confirmed, while on the other hand it the expression of the GluN1 subunit only has also been indicated [8,9]. The roles of NMDARs in peripheral non-neural tissues remain elusive due to differences in glutamate concentrations, the different composition of subunits where the NMDARs were built up and consequent modifications of NMDAR functionality, and variations in affinities for glutamate and glycine binding. The overactivation of NMDARs by homocysteine (Hcy) and subsequent pathological changes in the cardiovascular system have further illuminated the link between NMDAR function and the cardiovascular system [10,11].

Cardiovascular diseases, most notably ischemic heart disease, remain a major cause of death and disability worldwide. More than three decades ago the phenomenon of ischemic preconditioning provided new possibilities for the prevention and decrease of heart damage due to myocardial infarction [12]. As well as the model of ischemic preconditioning, consisting of short periods of ischemia and reperfusion prior to prolonged ischemia, other pharmacological and non-pharmacological approaches of heart conditioning have evolved [13,14]. Pharmacological pre- and postconditioning aim to activate protective cellular pathways through the activation of receptors and intracellular signaling cascades which favor survival of cardiac cells [15].

One of the most prominent pathological mechanisms which mediate cell damage due to ischemia and reperfusion is oxidative stress. Namely, the impairment of mitochondrial function and the depletion of ATP, combined with disturbances in Ca^2+^ content induced by ischemia and reperfusion, lead to an increased production of various reactive species, such as the superoxide anion radical (O_2^−^_), the hydroxyl radical (OH^−^), and peroxynitrite or hydrogen peroxide (H_2_O_2_), and consequent oxidative damage and cell death [16,17,18]. Impaired Ca^2+^ handling augments oxidative burst upon the restoration of the coronary flow and oxygen supply. Thus, contradictorily, re-establishing the blood flow exacerbates tissue damage [17,19]. On the other hand, it has been shown that ischemia and reperfusion increase the myocardial level of glutamate [20]. Furthermore, NMDAR-mediated Ca^2+^ influx in ischemic conditions and during reperfusion worsens myocardial damage by myocardial necrosis and the induction of cardiomyocyte apoptosis [21]. The application of MK-801, as a NMDAR blocker, or heart perfusion with Ca^2+^-free buffer diminishes these adverse effects of heart NMDAR overactivation. Given that the heart’s function is highly dependent on Ca^2+^ homeostasis, it has been shown that the acute application of NMDAR antagonists induces changes in the heart function and oxidative stress biomarkers outflow. The intracoronary administration of MK-801, memantine and ifenprodil decreases the production of reactive oxygen species and alters heart function, while the application of NMDA and homocysteine in combination with glycine also affects heart function and redox balance [22,23]. Furthermore, the simultaneous application of glutamate and glycine, as NMDAR’s co-agonists, affect the effects of verapamil as a Ca^2+^ channel blocker on isolated rat heart [24].

Given all the above mentioned facts, the aim of this study was to assess the effects of the preconditioning and postconditioning of isolated rat heart with NMDAR agonists (glutamate and (RS)-(Tetrazol-5-yl)glycine) and antagonists (memantine and MK-801) on heart function and the release of oxidative stress biomarkers.

## 2. Materials and Methods

### 2.1. Preparation of Isolated Heart

The hearts of male Wistar albino rats (8 weeks old, body mass index 200 ± 30 g, n = 120, 12 in each experimental group) were excised and perfused using a Langendorff apparatus (Experimetria Ltd., 1062 Budapest, Hungary). The sample size was calculated using software G*Power, according to previously published studies of similar methodological design [13,21,22,23]. The rats were obtained from the Military Medical Academy, Belgrade, Serbia and were kept on an artificial 12-h light–dark cycle (8:00 a.m.–8:00 p.m.) at room temperature (22 ± 2 °C). The animals were housed in their respective groups in a collective cage and received water and standard laboratory chow ad libitum.

After short-term inhalation ether anesthesia (ketamine was not used in order to avoid any effect on the hearts, due to its action via the NMDARs), the rats were sacrificed by cervical dislocation (Schedule 1 of the Animals/Scientific Procedures, Act 1986, UK). Short-term ether anesthesia was chosen in sacrificing the animal to minimize the effects of anesthetics on heart function. Given that diethyl ether possess less cardiodepressant action compared to pentobarbital, we opted for the use of ether [25]. Following urgent thoracotomy, the hearts were promptly excised and attached to the Langendorff apparatus via aortic cannulation. After aorta cannulation, the hearts were retrogradely perfused under a constant perfusion pressure (CPP) of 70 cmH_2_O with complex Krebs—Henseleit solution (KHS), composed of the following (in mmol/L): NaCl 118, KCl 4.7, CaCl_2_ × 2H2O 2.5, MgSO_4_ × 7H_2_O 1.7, NaHCO_3_ 25, KH_2_PO_4_ 1.2, glucose 11, pyruvate 2, equilibrated with 95% O_2_ plus 5% CO_2_, warmed to 37 °C (pH 7.4). Following the restoration of normal rhythm, a sensor (transducer BS4 73-0184, Experimetria Ltd., Budapest, Hungary) was placed into the left ventricle through an incision in the left atrium adjacent to a severed mitral valve for the continuous monitoring of cardiac function.

### 2.2. Experimental Protocol

All experimental groups underwent 25 min of perfusion at a coronary perfusion pressure (CPP) of 70 cm H_2_O (stabilisation period); during this period, each of the hearts was subjected to short-term occlusion (30s) followed by simultaneous bolus injections of 5 mmol/L adenosine (60 µL at a flow of 10 mL/min to elicit maximal coronary flow) to test coronary vascular reactivity. If coronary flow (CF) did not increase by 100% compared with control values, the hearts were discarded. Coronary flow was determined flowmetrically. Once CF was stabilized (three measurements of the same value), samples of coronary effluent were collected, and the experimental protocol was initiated. In the control group, all isolated rat hearts were subjected to global ischemia after the stabilization period (the perfusion of the whole heart with KHS was stopped) for 20 min, followed by 30 min of reperfusion. In the preconditioning (PreC) control group, we applied saline 5 min before ischemia, while in the postconditioning (PostC) group we applied saline during the first three minutes of reperfusion. In the PreC groups, after stabilization period, the hearts were perfused with the NMDAR agonists glutamate (100 µmol/L) and (RS)-(Tetrazol-5-yl)glycine (TG) (5 μmol/L) and the NMDAR antagonists memantine (100 μmol/L) and MK-801 (30 μmol/L), for 5 min, before global ischemia of 20 min which was followed by 30 min reperfusion. In the PostC group, the hearts were perfused with glutamate (100 µmol/L), TG (5 μmol/L), and the NMDAR antagonists memantine (100 μmol/L) and MK-801 (30 μmol/L) during the first 3 min of reperfusion.

Experimental groups in this study were:Control group preconditioned with saline (n = 12)Control group postconditioned with saline (n = 12)Group preconditioned with glutamate in dose of 100 µmol/L (n = 12)Group postconditioned with glutamate in dose of 100 µmol/L (n = 12)Group preconditioned with (RS)-(Tetrazol-5-yl)glycine in dose of 5 µmol/L (n = 12)Group postconditioned with (RS)-(Tetrazol-5-yl)glycine in dose of 5 µmol/L (n = 12)Group preconditioned with MK-801 in dose of 30 µmol/L (n = 12)Group postconditioned with MK-801 in dose of 30 µmol/L (n = 12)Group preconditioned with memantine in dose of 100 µmol/L (n = 12)Group postconditioned with memantine in dose of 100 µmol/L (n = 12)

The concentrations of the administered compounds were determined and modified according to previously published studies of similar experimental design conducted by others and ourselves [22,23,24,26,27,28]. The effects of two agonists and two antagonists of NMDARs were examined in this study. Glutamate was used as a common agonist of ionotropic glutamate receptors and (RS)-(Tetrazol-5-yl)glycine as a highly potent agonist of NMDARs. On the other hand, two antagonists of NMDARs were applied: memantine, a commonly accepted drug in the treatment of Alzheimer’s disease whose action is based on the inhibition of NMDARs, and MK-801, one of the most commonly used NMDAR antagonists in experimental studies. Memantine and MK-801 have a different mechanism of action [1].

Samples of coronary venous effluent were collected during the experiments at the same points of interest: after the stabilization period (C), after the application of the drug for PreC group, at the first, third, and fifth minutes of reperfusion, and in further intervals of 5 min until the end of the experiment. In the PostC group, venous effluent was collected after the stabilization period (C), at the first and third minutes (the last minute of the application of the drug), at the fifth minute of reperfusion, and in further intervals of 5 min until the end of the experiment. The following parameters of myocardial function were determined:The maximum rate of pressure development in the left ventricle (dp/dt max)The minimum rate of pressure development in the left ventricle (dp/dt mix)The systolic left ventricular pressure (SLVP)The diastolic left ventricular pressure (DLVP)The heart rate (HR)

### 2.3. Biochemical Assays

The following oxidative stress parameters were determined spectrophotometrically (Specord S-600 Analytik Jena) using collected samples of the coronary venous effluent:The index of lipid peroxidation, measured as thiobarbituric acid reactive substances (TBARS)The level of the superoxide anion radical (O_2^−^_)The level of hydrogen peroxide (H_2_O_2_)The level of nitrite (NO_2^−^_)

#### 2.3.1. TBARS Determination (Index of Lipid Peroxidation)

The degree of lipid peroxidation in the coronary venous effluent was estimated by measuring TBARS, using 1% thiobarbituric acid (TBA) in 0.05 NaOH, incubated with the coronary effluent at 100 °C for 15 min and measured at 530 nm. KHS was used as a blank probe, as previously described [29].

#### 2.3.2. Determination of the Hydrogen Peroxide Level

The measurement of the level of hydrogen peroxide (H_2_O_2_) was based on the oxidation of phenol red by hydrogen peroxide in a reaction catalyzed by horseradish peroxidase (HRPO), as previously described by Pick et al., and reviewed in [29]. An amount of 200 µL of perfusate was precipitated using 800 mL of freshly prepared phenol red solution; 10 µL of (1:20) HRPO (made ex tempore) was subsequently added. For the blank probe, an adequate volume of KHS was used instead of coronary venous effluent. The level of H_2_O_2_ was measured at 610 nm.

#### 2.3.3. Determination of the Nitrite Level

Nitric oxide decomposes rapidly to form stable nitrite/nitrate products. The nitrite level (NO_2^−^_) was measured spectrophotometrically and used as an index of nitric oxide (NO) production, using the Griess’s reagent as previously described by Green and coworkers, as previously described [29]. A total of 0.5 mL of perfusate was precipitated with 200 µL of 30% sulphosalicylic acid, vortexed for 30 min, and centrifuged at 3000× *g*. Equal volumes of the supernatant and Griess’s reagent, containing 1% sulphanilamide in 5% phosphoric acid/0.1% naphthalene ethylenediamine-di hydrochloride, were added and incubated for 10 min in the dark and measured at a wavelength of 543 nm. The nitrite levels were calculated using sodium nitrite as the standard.

#### 2.3.4. Determination of the Level of the Superoxide Anion Radical

The level of the superoxide anion radical (O_2^−^_) was measured via a nitro blue tetrazolium reaction in TRIS (tris(hydroxymethyl)aminomethane) buffer with coronary venous effluent at 530 nm, as previously described by Auclair et al., as previously described [29]. KHS was used as a blank probe.

### 2.4. Ethical Approval

All research procedures were carried out in accordance with the European Directive for the welfare of laboratory animals no. 86/609/EEC and the principles of good laboratory practice (GLP), approved by the Ethical Committee for Laboratory Animal Welfare of the Faculty of Medical Sciences, University of Kragujevac, Serbia (approval code: 01-13786/2, date: 12 December 2019).

### 2.5. Drugs

All drugs used in this experimental protocol were provided by Sigma-Aldrich.

### 2.6. Statistical Analysis

In the PreC group, we compared the last minute of stabilization (C), the last minute of application of the drug, and the last minute of the reperfusion period. In the PostC group, three points of interest were the last minute of stabilization (C), the third minute (the last minute of application of the drug), and the last minute of reperfusion. Values are expressed as mean ± standard error (SE). Statistical analysis was performed by a nonparametric matched-pair signed-rank test (Wilcoxon). *p* values lower than 0.05 were considered to be significant.

## 3. Results

### 3.1. Effects of NMDA Conditioning on Cardiodynamic Parameters and Coronary Flow in Isolated Rat Heart

#### 3.1.1. The Effects of NMDAR Conditioning with Glutamate and TG on the Cardiodynamic Parameters and Coronary Flow in Isolated Rat Heart

In the preC control group, the values of all cardiodynamic parameters, except DLVP, were significantly lower in the last minute of reperfusion compared to the initial values (Figure 1A,E, Figure 2A,E, and Figure 3A,E). In the PostC control group, dp/dt max, dp/dt min and SLVP were significantly increased in the third minute of reperfusion compared to the incipient values (Figure 1C,G and Figure 2C), while the values of HR and CF were lower (Figure 3C,G). At the last minute of reperfusion, dp/dt max, dp/dt min, SLVP, HR, and CF were significantly decreased compared to the third minute of reperfusion and the initial values (Figure 1C,G, Figure 2C, and Figure 3C,G).

In the PreC glutamate group, the acute application of glutamate did not induce change in any cardiodynamic parameter. During the reperfusion period all parameters, except SLVP and DLVP, decreased and reached values significantly lower in relation to the initial and last minutes of glutamate application (Figure 1A,E and Figure 3A,E).

In the PostC glutamate group, the values of all measured cardiodynamic parameters, except DLVP, were significantly lower in the last minute of reperfusion compared to the initial values (Figure 1C,G, Figure 2C, and Figure 3C,G). HR was lower in the third minute of reperfusion compared to the initial value of HR, and this decreasing trend continued until the end of reperfusion (Figure 3C).

In the PreC TG group, the acute application of TG induced a significant decrease in dp/dt max, dp/dt min, and HR (Figure 1A,E and Figure 3A), while SLVP was increased (Figure 2A). In the last minute of reperfusion, the values of dp/dt max, HR, and CF were significantly lower in relation to the initial values (Figure 1A and Figure 3A,E).

In the PostC TG group, postconditioning with TG significantly increased the values of SLVP and DLVP in the third minute of reperfusion (the last minute of TG application) in comparison to the initial values (Figure 2C,G), while HR was significantly lower (Figure 3G). The dp/dt max, HR, and CF values were lower at the end of reperfusion in comparison to the initial values (Figure 1C and Figure 3C,G).

Taken altogether, neither glutamate nor TG had a protective effect on heart function. Most of the measured cardiodynamic parameters were significantly lower in the last minute of reperfusion compared to the initial values. These results suggest the assumption that the activation of cardiac NMDARs before or after ischemia has adverse effects on cardiac function.

#### 3.1.2. The Effects of NMDAR Conditioning with Memantine and MK-801 on Cardiodynamic Parameters and Coronary Flow in Isolated Rat Heart

In the PreC memantine group, the acute application of memantine induced a significant decrease in all cardiodynamic parameters, excluding DLVP (Figure 1B,F, Figure 2B, and Figure 3B,F). During the reperfusion period dp/dt max and dp/dt max significantly increased, but did not reach values similar to the initial values (Figure 1B,F). Furthermore, HR and CF were lower at the end of reperfusion in comparison to their initial values (Figure 3B,F).

In the PostC memantine group, the HR value was lower in the third minute of reperfusion (last minute of postconditioning with memantine) in relation to the initial value (Figure 3D). The values of dp/dt min, HR, and CF significantly decreased during reperfusion in comparison to the incipient values (Figure 1H and Figure 3D,H).

In the PreC MK-801 group, the acute application of MK-801 induced a significant decrease in all observed cardiodynamic parameters (Figure 1B,F, Figure 2B,F, and Figure 3B,F). During the reperfusion all parameters were significantly increased and only HR and CF did not reach values similar to the initial values (Figure 3B,F).

In the PostC MK-801 group, the administration of MK-801 in postconditioning during the first three minutes of reperfusion induced a significant decrease in dp/dt max, dp/dt min, SLVP, and HR, while CF was increased (Figure 1D,H, Figure 2D, and Figure 3D,H). During reperfusion all parameters reached values similar to the initial values (Figure 1D,H, Figure 2D, and Figure 3D,H).

Contrary to the effects of NMDAR agonists, memantine and MK-801 exhibited better protective properties on cardiac function. The most powerful protective effect was when MK-801 was applied during PostC, given that all cardiodynamic parameters at the end of reperfusion reached values similar to the initial values.

### 3.2. Effects of NMDA Conditioning on Biomarkers of the Oxidative Stress in Isolated Rat Heart

#### 3.2.1. The Effects of NMDAR Conditioning with Glutamate and TG on the Biomarkers of Oxidative Stress in Isolated Rat Heart

In the preC control group, the values of TBARS, H_2_O_2_ and O_2_^−^ were significantly increased in the last minute of reperfusion compared to the initial values (Figure 4A,E and Figure 5E). In the PostC control group, TBARS and O_2_^−^ were significantly higher in the third minute of reperfusion compared to their initial values (Figure 4C and Figure 5G). The values of TBARS, H_2_O_2_, and O_2_^−^ at the last minute of reperfusion were significantly increased in comparison to the control point (Figure 4C,G and Figure 5G).

In the preC glutamate group, only O_2_^−^ was significantly increased at the last minute of reperfusion compared to the control values (Figure 5E). In the PostC glutamate group, the values of O_2_^−^ were increased in the third minute of reperfusion (the last minute of postconditioning with glutamate) in comparison to the incipient values (Figure 5G). In the last minute of reperfusion TBARS and O_2_^−^ were significantly higher in comparison to the initial values (Figure 4C and Figure 5G).

In the preC TG group, the acute application of TG induced a significant increase of O_2_^−^ in relation to the initial values (Figure 5E). In the last minute of reperfusion TBARS and O_2_^−^ were significantly increased compared to the incipient values (Figure 4A and Figure 5E). In the PostC TG group, only O_2_^−^ was significantly higher in the last minute of reperfusion compared to its initial value (Figure 5G).

The NMDAR agonists had prooxidative effects, affecting mostly O_2_^−^, resulting in an increase in TBARS in the last minute of reperfusion. These results suggest that increased O_2_^−^ upon the activation of NMDARs and consequent molecular and cellular damage could be the one of the mechanisms that mediate the reduction of cardiac function.

#### 3.2.2. The Effects of NMDAR Conditioning with Memantine and MK-801 on the Biomarkers of Oxidative Stress in Isolated Rat Heart

In the PreC memantine group, the acute administration of memantine significantly decreased the values of NO_2_^−^ and O_2_^−^ in relation to their initial values (Figure 5B,F). In the last minute of reperfusion all oxidative stress biomarkers, except O_2_^−^, were significantly lower in comparison to the incipient values (Figure 4B,F and Figure 5B,F).

In the PostC memantine group, the values of H_2_O_2_ and NO_2_^−^ were significantly lower in the third minute of reperfusion (the last minute of postconditioning with memantine) in relation to the start values, and these two parameters remained significantly lower in relation to the incipient values (Figure 4H and Figure 5D).

In the PreC MK-801 group, all oxidative stress biomarkers significantly decreased during preconditioning with MK-801 (Figure 4B,F and Figure 5B,F). In the last minute of reperfusion H_2_O_2_ and NO_2_^−^ were significantly lower compared to the initial values (Figure 4F and Figure 5B).

In the PostC MK-801 group, the value of H_2_O_2_ was significantly lower in the last minute of reperfusion compared to the start value (Figure 4H).

Both applied NMDAR antagonists, memantine and MK-801, exerted antioxidative properties. Such antioxidative effects of memantine and MK-801 combined with better cardiac function, and the prooxidative effects of NMDAR agonists confirm the assumption of the importance of oxidative stress as an etiological factor in ischemia and reperfusion of the heart, as well as NMDARs as factors involved in its regulation.

## 4. Discussion

The aim of this study was to assess the role of activation and inhibition of cardiac NMDARs, applied by preconditioning and postconditioning, on changes in cardiodynamic parameters and oxidative stress biomarkers during ischemia and reperfusion in a model of isolated rat heart.

In our previous experiments, the acute application of glutamate or NMDA alone did not change parameters of cardiodynamic parameters, while the combined application of glutamate and glycine altered cardiac contractility [22,24,31]. These variations in heart contractility could be related to the different sensitivity of heart NMDARs to glutamate and glycine, as well as the experimental technique, given that Langendorff retrograde perfusion of isolated heart implies an ex vivo technique, where substances not found in KHS are gradually washed out of the cardiac tissue. Moshal et al. indicated the link between Hcy and cardiac NMDARs [32]. The activation of NMDARs in the heart by Hcy leads to disturbances in Ca^2+^ handling due to changes in the calcium-handling proteins SERCA 2a and NCX (sodium/calcium exchanger). Furthermore, increased intracellular levels of Ca^2+^ induce the activation of matrix metalloproteinase (MMP) and impaired mitochondrial function, and as a final result contractile dysfunction develops. The role of NMDARs in Hcy-induced contractile dysfunction was confirmed by the use of NMDAR blockers whereby all deleterious effects were attenuated. The acute application of DL-Hcy thiolactone, in a similar experimental protocol, also decreased cardiac contractility [33,34]. Compared to preconditioning, postconditioning with NMDAR agonists had a less deleterious effect on cardiac contractility. In patients undergoing coronary artery bypass surgery, the intraoperative and postoperative intravenous infusion of glutamate significantly decreased postoperative levels of N-terminal pro-B-type natriuretic peptide (NT-proBNP) in high-risk patients, showing that glutamate may preclude or reduce myocardial dysfunction [35]. However, it was also shown that the glutamate level in tissues affected by ischemia rises [20]. Regardless of the role of glutamine as a transmitter, it represents a very important amino acid in cellular metabolism, especially in ischemic conditions and depleted oxygen consumption [36]. This double-edged sword effect of glutamate, both protective and adverse, seems to be dependent on the time window of the glutamate increase, the effect on glutamate receptors, and the consequent Ca^2+^ influx [37].

In our previous study, the acute application of MK-801 also decreased cardiac contractility [23]. Jannesar et al. assessed the effects of memantine on heart function in an ex vivo model of ischemia/reperfusion of isolated rat heart, as well as in an in vivo model of isoproterenol-induced myocardial infarction [38]. The results of their study showed that the application of memantine for seven consecutive days prior to global ischemia and reperfusion significantly improved the parameters of cardiac contractility and left ventricular systolic pressure combined with a reduction of infarct size. Furthermore, the in vivo part of this study also indicated the protective role of memantine due to the reduction of ECG abnormalities induced by isoproterenol and the improvement of mean arterial pressure. The beneficial effects of memantine in heart ischemia and myocardial infarction were mediated by a decrease in oxidative stress and a reduction in inflammation [38]. Other studies dealing with the role of NMDAR antagonists in conditioning have mostly focused on the nervous system. The pretreatment of primary cultures of the cerebellar granule cells with MK-801 or memantine was found to minimize the deleterious effects of deprivation of glucose and oxygen, as well as the excitotoxic levels of glutamate [39]. MK-801 and memantine reduced Ca^2+^ influx, probably due to the prevention of NMDAR activation, which reflected the activation of protective signaling pathways and/or the inactivation of signaling cascades leading to apoptosis or necrosis. The results of previous studies also indicated the protective effects of preconditioning with NMDA antagonists in neurons treated with high doses of NMDA, α-amino-3-hydroxy-5-methyl-4-isoxazolepropionic acid (AMPA), staurosporine, etoposide, or oxygen/glucose deprivation [40]. One of the suggested mechanisms of neuroprotection includes the prevention of activation of protein kinase C (PKC) due to NMDA-induced Ca^2+^ influx, and subsequent events such as calpain activation. Interaction between calpain proteases, intracellular Ca^2+^, and PKC appears to be affected in ischemic myocardium and play an important role in cell shifting between programmed cell death and cell survival [41,42]. Taken altogether, it appears to be that similar mechanisms regarding the relation between NMDARs and ischemic/reperfusion (I/R) damage may be affected both in cardiomyocytes and neurons. Some future research on assessing the place of cardiac NMDARs in heart homeostasis should be directed to clarifying their role in the orchestration of different signaling pathways in I/R.

Similar to previously published results, SLVP and DLVP did not change during glutamate administration [31], while the activation of NMDARs by TG induced pressor response. Most studies dealing with the effects of NMDAR and blood pressure are related to the modulation of central NMDARs and consequent pressor response [43]. McGee and Abdel-Rahman showed that NMDA-induced pressor response in conscious rats could be prevented by the inhibition of phosphoinositide 3-kinase (PI3K)-Akt, protein kinase C, Ca^2+^ influx, or NADPH oxidase [44]. The results of this study confirmed that signaling pathways including NMDAR, PI3K-Akt, PKC and NADPH oxidase (NOX) mediate in changes of blood pressure. The regulation of Ca^2+^ in many ways arises as one of the crucial aspects in signaling cascades due to NMDAR activity. Similarly to the results of this study, it was shown that preconditioning with memantine improved left ventricle systolic pressure, but did not affect end-diastolic pressure [38]. The effects of NMDAR modulation obtained in isolated heart probably arose as a consequence of changes in Ca^2+^ handling in heart tissues. It was shown that deprivation of oxygen and glucose in human neonatal cardiomyocytes resulted in increased NMDAR activity and a consequently enhanced calcium influx [45]. Increase in Ca^2+^ inflow was accompanied by the activation of p38 mitogen-activated protein kinase (MAPK), and consequent apoptosis and decreased cell viability. These deleterious effects were abolished by application of the NMDAR blocker MK-801. The results of this study, at least to some extent, confirm the key role of Ca^2+^ in changes induced by different NMDAR activities in the heart. Future investigations should deal with possible molecular cascades and signaling pathways which interfere in NMDAR-mediated changes of heart function.

In our previous studies, both MK-801 and memantine induced a decrease of HR and CF, which is in accordance with results of this study [22,23]. Shi et al. examined the effects of the chronic activation of NMDARs on cardiac electrophysiology and heart rate [46]. The results of this study show that chronic NMDAR stimulation induce an increase in HR measured in vivo and prolongation in QT, QTc, and TpTe intervals. Furthermore, these changes were accompanied with alterations in the expression of some potassium channel proteins. On the other hand, the application of MK-801 mitigated such cardiac electrical remodeling induced by chronic NMDA treatment. Such changes in the electrical currents of the heart may have important roles in arrhythmogenesis, especially in conditions with a pronounced tendency towards arrhythmias, such as ischemia. By assessing the role of NMDARs in reperfusion-induced arrhythmias, it was shown that the application of MK-801 significantly reduced ventricular arrhythmia scores [47]. Furthermore, a possible mechanism arose relating to the increase in sarcoplasmic reticulum Ca^2+^-ATPase activity induced by MK-801, as well as the increased expression of the SERCA (sarco/endoplasmic reticulum Ca^2+^-ATPase) 2a protein. Improved regulation of intracellular Ca^2+^ by MK-801 was followed by a decrease in intramitochondrial Ca^2+^, suggesting better maintenance of ATP production and a delay in ATP depletion. The results of the research conducted to examine the effects of NMDAR ionotropic channel antagonists in ischemia and I/R induced arrhythmias indicated that the blockage of NMDARs by MK-801 or memantine prevented arrhythmias if they are applied in reperfusion [48]. It can be assumed that these results correlate with the results of this study, given that protective effects were found in group postconditioned with MK-801, as a nonselective NMDAR antagonist (Figure 3D). It was also shown that MK-801 increases heart rate variability (HRV), an index that reflects the balance of autonomous regulation of heart function, and thus decreases the probability of arrhythmias occurring [49]. In accordance with the mentioned results, the chronic activation of NMDARs through daily injections of NMDA resulted in increased HRV, increased atrial electrical remodeling, and atrial fibrosis [50]. The coadministration of MK-801 with NMDA resulted in the moderation of those alterations.

CF, which reflects coronary blood vessel functionality, was best preserved in the group postconditioned with MK-801 (Figure 3H). The reactivity of coronary blood vessels remains one of the most important physiological properties of the heart. In an experimental setup designed to assess the effects of asphyxia in fetal sheep induced by umbilical cord occlusion, Dean et al. showed that the increase in vascular resistance was reduced by MK-801 [51]. The role of NMDARs in the regulation of blood flow has been mainly examined in relation to the cerebral blood flow. In physiological, normoxic conditions the activation of NMDARs induces the activation of nitric oxide synthase (NOS) and a consequent increase in nitric oxide (NO) production which results in vasodilatation [52]. Another pathway involved in NMDAR-mediated vasodilatation implies the activity of the cytochrome P-450 epoxygenase. As well as the beneficial role of NO in the regulation of vascular reactivity in a prooxidant environment such as I/R, the inducible form of NOS (iNOS) is activated leading to increased production of O_2_^−^ [53]. Due to this fact it was shown that MK-801 decrease I/R injury of sciatic nerve by decrease of oxidative damage and activity of iNOS [54]. Furthermore, given that vascular smooth muscle cells (VSMC) also express NMDARs, it was shown that Hcy, as an NMDAR activator, may induce the production of C-reactive protein (CRP) and oxidative stress, while MK-801 prevented such changes [55]. Namely, the increased activity of NMDARs due to the resence of Hcy induced an increase in the production of reactive oxygen species (ROS), which in turn caused the activation of mitogen-activated protein kinase (MAPK) and nuclear factor κB (NF-κB). Thus, the creation of the signaling cascade NMDARs-ROS-MAPK-NF-κB leads to increased levels of CRP and inflammation. The facts mentioned indicate the possibility of interconnection between the activity of NMDARs in coronary arteries, Ca^2+^ handling, and the production of various substances, which, altogether, may change the reactivity of coronary blood vessels. One future research direction may be the role of NMDARs in the reactivity of coronary circulation, with attention focused on the localization of NMDARs (endothelial cells or VSMC) and the signaling molecules involved.

Another part of the experimental protocol was dedicated to the assessment of the role of NMDAR activity in the production of oxidative species and oxidative stress. The preliminary results regarding the effects of MK-801 on oxidative stress biomarkers have been partially published [30]. One of the major mechanisms mediating in the deleterious effects of ischemia and reperfusion is oxidative stress. The main point of myocardial ischemic injury represents Ca^2+^ overload, which worsens during reperfusion. On the other hand, the sudden oxygen supply in reperfusion in combination with the disturbed Ca^2+^ load leads to oxidative burst and oxidative damage of the cell structures [17]. Given the function of NMDARs as a Ca^2+^ channel, as well as the modulation of their activity in I/R, their contribution to redox balance in the heart could be of considerable importance. The above mentioned study by McGee and Abdel-Rahman indicated the mediation of PI3K, PKC, and NOX in increased production of reactive species in vasculature by NMDAR activation [43]. Gao et al. showed that the treatment of isolated cardiomyocytes with NMDA results in increased Ca^2+^ influx and the generation of ROS [28]. Furthermore, NMDA induced an increase in cytochrome c level in the cytoplasm of cardiomyocytes, and activation of caspase-3, which consequently decreased cardiomyocyte viability and increased apoptosis. All these deleterious consequences of NMDAR activation were reduced by MK-801. Our prior results also showed the antioxidative effects of MK-801 [23]. It has also been shown that blocking peripheral NMDARs by ethanol induces a decrease in the production of reactive species in the cardiovascular system induced by NMDA infusion [56]. Some characteristics and effects of the interaction between ethanol and NMDARs in the CNS are known [57], but the possibility of modulation of vascular NMDARs by ethanol is still elucidative. The effects of memantine on redox balance have been mainly examined in neural structures, given that memantine is used in therapy for Alzheimer’s disease (AD). Namely, the level of oxidative stress in AD patients is increased, while therapy with memantine exhibits a tendency to decrease oxidative damage [58]. The protective effects of memantine in an ex vivo model of global ischemia in isolated rat heart and an in vivo model of isoproterenol-induced myocardial infarction were affected by the reduction of cardiac oxidative stress and myocardial inflammatory reaction [37]. The application of memantine induced the reduction of the myocardial malondialdehyde (MDA) level, as a marker of oxidative damage, and myocardial myeloperoxidase activity, as a marker of neutrophil infiltration, reflecting myocardial inflammation. These results correlate with the results of this study, confirming the association between NMDAR activity and oxidative stress in heart tissue, and Ca^2+^ as a link between them. In a study assessing the effects of methylmercury-induced toxicity in rats, memantine showed pronounced protective properties [59]. Methylmercury is a common pollutant which induces misbalance in glutamate signaling and oxidative stress. The application of memantine prior to the methylmercury exposure reduced NMDAR activation and decreased Ca^2+^ influx and the production of ROS in the cerebral cortex. Memantine also improved the enzymatic antioxidative defense and oxidative damage of the neurons. Relying on the results of this study and the conclusions of other research, NMDARs could be one of the crucial mediators in the regulation of oxidative balance in many tissues. In future investigations, attention should be paid to NMDARs in the cardiovascular system and their role in the regulation of oxido-reduction processes during ischemia and reperfusion. Given that perioperative myocardial infarction represents a leading cause of mortality due to cardiac surgery interventions [60], the modulation of cardiac NMDARs could be a new, promising cardioprotective tool in the battle for better survival and prognosis in patients undergoing non-cardiac and cardiac surgical procedures.

## 5. Conclusions

The results of this study show that cardiac NMDARs may be a therapeutic focus in ischemia and reperfusion injury. The application of NMDAR antagonists, especially MK-801, at the beginning of reperfusion exhibit the most pronounced protective effect. This protection is undoubtedly mediated by alterations in redox balance and the production of ROS. Future studies in this field should be directed to the creation of an NMDAR antagonist with an action specific to cardiac NMDARs. Furthermore, it is necessary to examine which time windows in ischemia and reperfusion are most ideal for the modulation of cardiac NMDARs in order to achieve the best therapeutic effect, maximum protection of the heart, and the preservation of reactivity of coronary circulation.

## Figures and Tables

**Figure 1 biomolecules-10-01065-f001:**
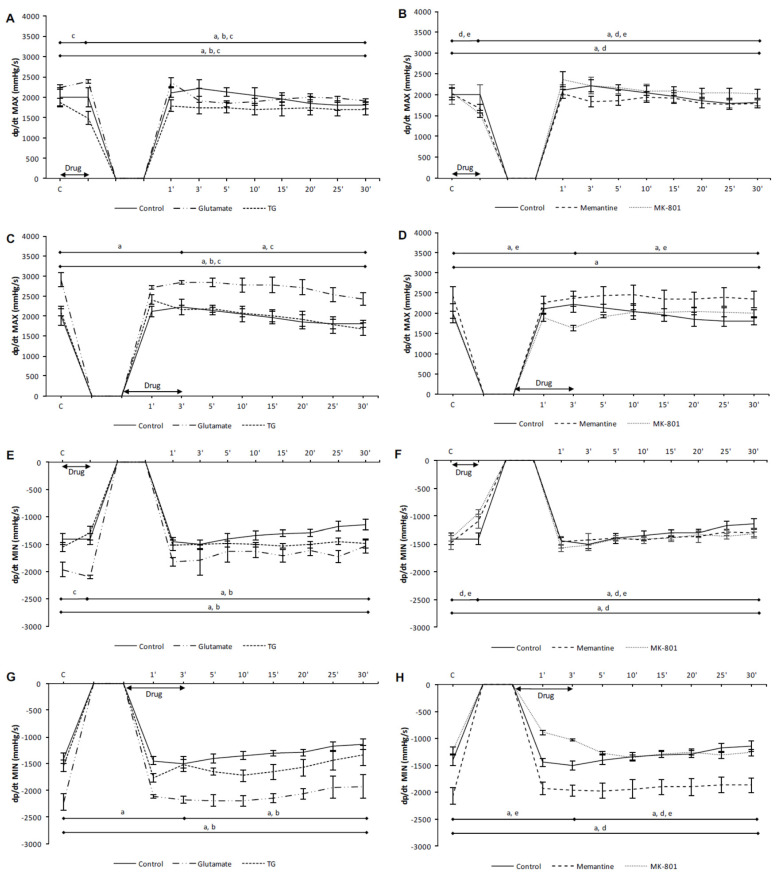
The effects of cardiac N-methyl-D-aspartate receptor (NMDAR) modulation in preC and postC on parameters of cardiac contractility. (**A**,**E**) preconditioned with glutamate and TG; (**B**,**F**) preconditioned with memantine and MK-801; (**C**,**G**) postconditioned with glutamate and TG; (**D**,**H**) postconditioned with memantine and MK-801. dp/dt max—maximum rate of pressure development in the left ventricle; dp/dt min—minimum rate of pressure development in the left ventricle; TG—(RS)-(Tetrazol-5-yl)glycine; preC—preconditioning; postC—postconditioning. Statistical significance between points of interest was presented as: a—in control group; b—in glutamate group; c—in TG group; d—in memantine group; e—in MK-801 group. Statistical significance was considered significant if the *p* value was less than 0.05 (*p* < 0.05).

**Figure 2 biomolecules-10-01065-f002:**
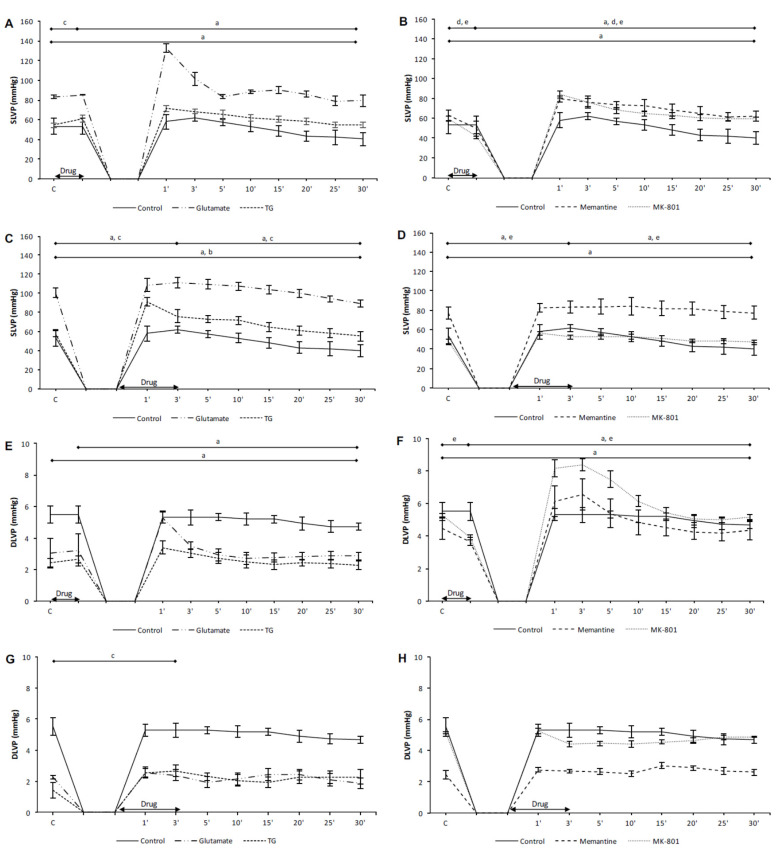
The effects of cardiac NMDAR modulation in preC and postC on systolic and diastolic pressure. (**A**,**E**) preconditioned with glutamate and TG; (**B**,**F**) preconditioned with memantine and MK-801; (**C**,**G**) postconditioned with glutamate and TG; (**D**,**H**) postconditioned with memantine and MK-801. SLVP—systolic left ventricular pressure; DLVP—diastolic left ventricular pressure; TG—(RS)-(Tetrazol-5-yl)glycine; preC—preconditioning; postC—postconditioning. Statistical significance between points of interest was presented as: a—in control group; b—in glutamate group; c—in TG group; d—in memantine group; e—in MK-801 group. Statistical significance was considered significant if the *p* value was less than 0.05 (*p* < 0.05).

**Figure 3 biomolecules-10-01065-f003:**
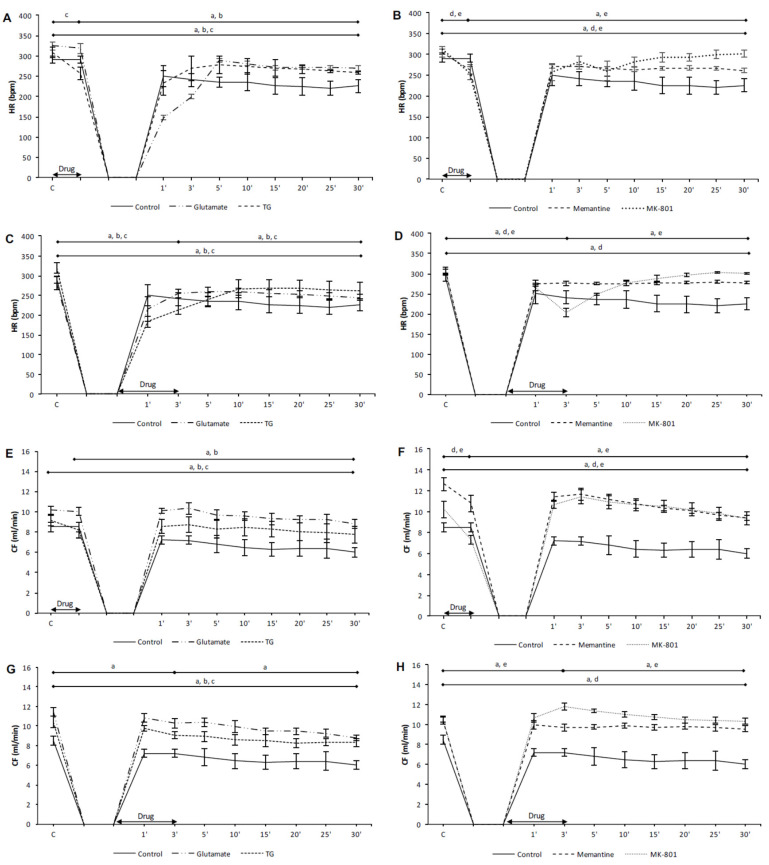
The effects of cardiac NMDAR modulation in preC and postC on heart rate and coronary flow. (**A**,**E**) preconditioned with glutamate and TG; (**B**,**F**) preconditioned with memantine and MK-801; (**C**,**G**) postconditioned with glutamate and TG; (**D**,**H**) postconditioned with memantine and MK-801. HR—heart rate; CF—coronary flow. TG—(RS)-(Tetrazol-5-yl)glycine; preC—preconditioning; postC—postconditioning. Statistical significance between points of interest was presented as: a—in control group; b—in glutamate group; c—in TG group; d—in memantine group; e—in MK-801 group. Statistical significance was considered significant if the *p* value was less than 0.05 (*p* < 0.05).

**Figure 4 biomolecules-10-01065-f004:**
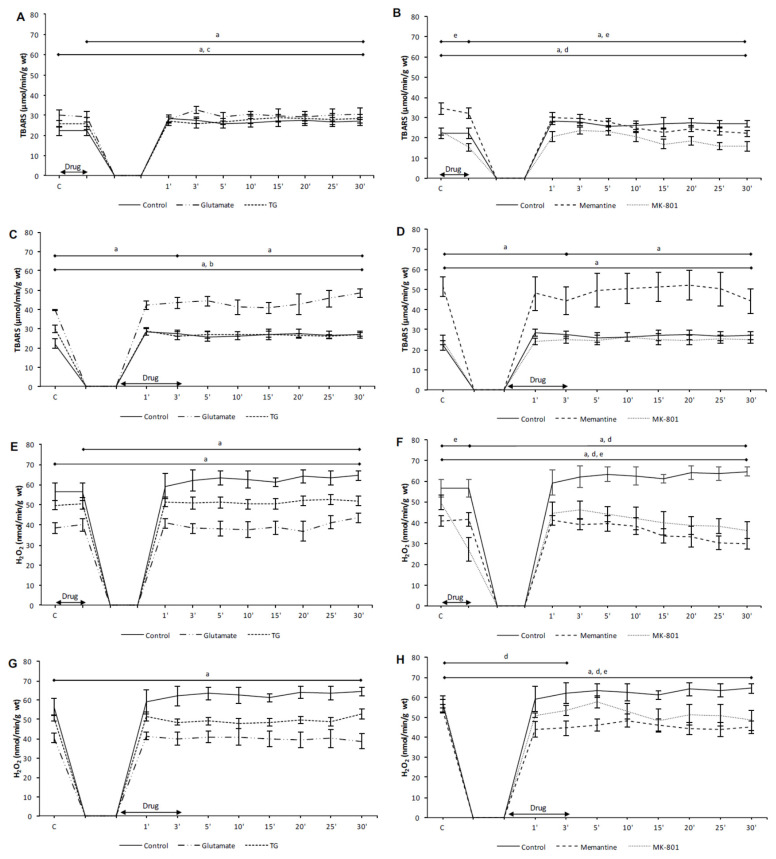
The effects of cardiac NMDAR modulation in preC and postC on TBARS and H_2_O_2_. (**A**,**E**) preconditioned with glutamate and TG; (**B**,**F**) preconditioned with memantine and MK-801; (**C**,**G**) postconditioned with glutamate and TG; (**D**,**H**) postconditioned with memantine and MK-801. TBARS—index of lipid peroxidation measured as TBARS; H_2_O_2_—hydrogen peroxide; TG—(RS)-(Tetrazol-5-yl)glycine; preC—preconditioning; postC—postconditioning. Some results regarding the effects of MK-801 on oxidative balance have been partially published [30] as a preliminary report. Statistical significance between points of interest was presented as: a—in control group; b—in glutamate group; c—in TG group; d—in memantine group; e—in MK-801 group. Statistical significance was considered significant if the *p* value was less than 0.05 (*p* < 0.05).

**Figure 5 biomolecules-10-01065-f005:**
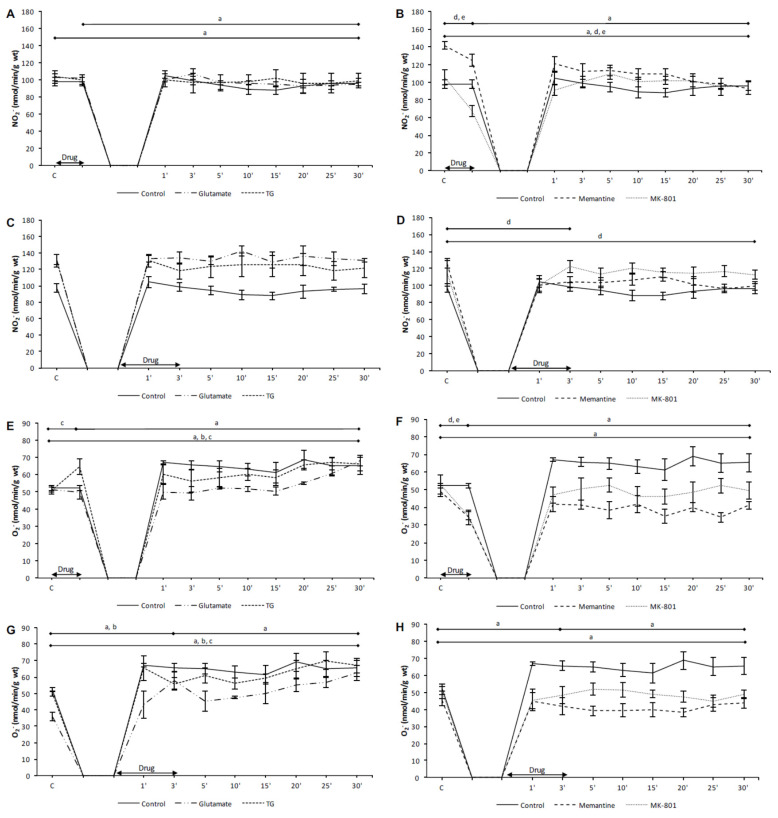
The effects of cardiac NMDAR modulation in preC and postC on NO_2_^−^ and O_2_^−^. (**A**,**E**) preconditioned with glutamate and TG; (**B**,**F**) preconditioned with memantine and MK-801; (**C**,**G**) postconditioned with glutamate and TG; (**D**,**H**) postconditioned with memantine and MK-801. NO_2_^−^—nitrites; O_2_^−^—superoxide anion radical; TG—(RS)-(Tetrazol-5-yl)glycine; preC—preconditioning; postC—postconditioning. Some results regarding the effects of MK-801 on oxidative balance have been partially published [30] as a preliminary report. Statistical significance between points of interest was presented as: a—in control group; b—in glutamate group; c—in TG group; d—in memantine group; e—in MK-801 group. Statistical significance was considered significant if the *p* value was less than 0.05 (*p* < 0.05).

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
