# Peer review of "The Role of Cardiac N-Methyl-D-Aspartate Receptors in Heart Conditioning—Effects on Heart Function and Oxidative Stress"

_biomolecules, 2020, doi:10.3390/biom10071065_

Round 1

Reviewer 1 Report

Govoruskina et al have performed an interesting ex vivo study in order define the pre or post-conditioning cardioprotective effects due to NMDARs agonists (glutamate 36 (100 μmol/l), (RS)-(Tetrazol-5-yl)glycine (5 μmol/l)) and antagonists (memantine 37 (100 μmol/l), MK-801 (30 μmol/l )) in male rat heart. Based on cardiac function and oxidative stress measurements, the authors have demonstrated that postconditioning by MK-801 is more effective in leading to cardioprotection. Some issues should be clarified.

1) Recent study has demonstrated that memantine, a NMDARs antagonist, exerts cardioprotective effects ex vivo by reducing pro-inflammatory and oxidative stress factors (please see PMID: 32544502). The authors should discuss their data in light of this recent unmentioned finding.

2) Previous unmentioned study has demonstrated that MX-801, which hampers calcium influx into cardiomyocytes, prevents ischemia-induced apoptosis of cardiomyocytes by limiting calcium-dependent p38 MAPK activity (please see PMID: 30614047). The authors should measure levels of p38 MAPK activity in order to better support the more effective cardioprotection by MK-801.

3) The authors should show significant differences (if any) in each graph. The use of tabled only to show statistical significance is confusing.

4) The authors should discuss their findings in light of translational perspectives involving perioperative cardioprotection (please see PMID: 31808661), which remains without effective solutions and your data could be helpful. 

Author Response

1) Recent study has demonstrated that memantine, a NMDARs antagonist, exerts cardioprotective effects ex vivo by reducing pro-inflammatory and oxidative stress factors (please see PMID: 32544502). The authors should discuss their data in light of this recent unmentioned finding.

The recent results regarding the cardioprotective effects of memantine are discussed and commented. Please see discussion section, line 456-465; 500-502 and 610-617.

2) Previous unmentioned study has demonstrated that MX-801, which hampers calcium influx into cardiomyocytes, prevents ischemia-induced apoptosis of cardiomyocytes by limiting calcium-dependent p38 MAPK activity (please see PMID: 30614047). The authors should measure levels of p38 MAPK activity in order to better support the more effective cardioprotection by MK-801.

Unfortunately, in this moment we are not able to assess the role of p38 MAPK activity due to NMDA receptor modulation in ischemic and reperfused heart within a reasonable period of time, especially having in mind the overall situation in the world. We corrected the omission and discussed the results of mentioned study in the discussion section, line 509-516. On the other hand, giving the interesting functional results presented in this study, we aim to investigate activity of several signaling pathways, including p38 MAPK, in order to understand deeper mechanisms that mediate the given effects.

3) The authors should show significant differences (if any) in each graph. The use of tabled only to show statistical significance is confusing.

Statistical significances are added in the graphs according to the reviewer suggestion. Please see the Figures 1-5. Tables with statistical significances are deleted from the manuscript. 

4) The authors should discuss their findings in light of translational perspectives involving perioperative cardioprotection (please see PMID: 31808661), which remains without effective solutions and your data could be helpful.

According to the reviewer’s suggestion, the results of the present study are discussed in relation to the translational perspectives and application in clinical practice. Please see discussion section, line 626-630.

Reviewer 2 Report

The study presented by Natalia Govoruskina and co-authors is focused on the pharmacological application of several substances with an aim to reach the cardioprotective effect after ischemia-reperfusion injury. The experimental design is focused on the NMDAR

 and contains several experimental groups and NMDAR agonists and antagonists. Overall the study represents a robust experimental work accompanied by physiological and biochemical studies. The study design is appropriate and the number of animals included in the group is sufficient for the precise statistical analysis.  Several minor correction can improve the ms:

  1. The discussion is too long and a bit unfocused simply representing the repetition of the results section, I would recommend to shorten it.
  2. Additionally, a short conclusion after every section in the Results summarising the most important findings of the section is recommended.
  3. The term “preconditioning” is usually used to apply a physiological or pharmacological stimuli to improve heart function and ischemia tolerance. There for in the current design the term “pharmacological intervention” or some others, but not “preconditioning” would be more appropriate.
  4. In describing the heart function, the arrhythmic phenotype should be mentioned or discussed. Were there any VT in postischemic period? Were they analysed?
  5. The significance is not stated in most of the figures, there for p-value must be marked somehow on the figures.

After all  the study represents a nice thorough experimental work accompanied by biochemical studies and can be published after minor corrections.

Author Response

1) The discussion is too long and a bit unfocused simply representing the repetition of the results section, I would recommend to shorten it.

The discussion section has been shortened according to the reviewer’s recommendation.

2) Additionally, a short conclusion after every section in the Results summarising the most important findings of the section is recommended.

Short conclusions have been added after every section in the results section.

3) The term “preconditioning” is usually used to apply a physiological or pharmacological stimuli to improve heart function and ischemia tolerance. There for in the current design the term “pharmacological intervention” or some others, but not “preconditioning” would be more appropriate.

All tested substances were applied both prior to ischemia, in order to assess the effects on heart function after ischemia and during reperfusion, and immediately after ischemia, in the first three minutes of reperfusion, so in that sense it could be considered as “preconditioning” and “postconditioning”. On the other hand we were not sure whether applied substances have protective or harmful effect, we wanted to investigate how agonists and antagonist of NMDARs affect cardiac function if administered before and after ischemia.

4) In describing the heart function, the arrhythmic phenotype should be mentioned or discussed. Were there any VT in postischemic period? Were they analysed?

The role of cardiac NMDA receptors in pathogenesis of postischemic arrhythmias could be very important side of their function in the heart. As we mentioned in the text, it is already shown that MK-801 has beneficial effects in ventricular arrhythmias reduction during reperfusion (reference number 46). Furthermore, the focus of next investigation regarding the cardiac NMDA receptors could be the incidence of arrhythmias during application of NMDA receptors agonists and antagonists. Also, the applied model of isolated heart was not appropriate for analysis of antiarrhythmic and arrhythmogenic effects of applied substances, giving the fact there was no technical possibility of pacing of the heart and the ECG during the experiment.

5) The significance is not stated in most of the figures, there for p-value must be marked somehow on the figures.

Statistical significances are added in the graphs according to the reviewer suggestion. Please see the Figures 1-5. Tables with statistical significances are deleted from the manuscript.